# Identification and Full-Length Sequence Analysis of a Novel Recombinant Goat Astrovirus Genotype in Guangxi, China

**DOI:** 10.3390/v16081213

**Published:** 2024-07-29

**Authors:** Yulu Zhou, Pengju Xu, Yanhua Huang, Jie Wang, Chang Cui, Yanglin Wang, Yuhang Luo, Xiaoling Wang, Jiang Xie, Fengmei Li, Zuzhang Wei, Ying Chen, Kang Ouyang, Yifeng Qin, Yan Pan, Weijian Huang

**Affiliations:** 1Laboratory of Animal Infectious Diseases and Molecular Immunology, College of Animal Science and Technology, Guangxi University, Nanning 530004, China; z205020639@163.com (Y.Z.); zuzhangwei@gxu.edu.cn (Z.W.); yingchen1@gxu.gxn.edu.cn (Y.C.);; 2Laboratory for the Prevention and Control of Bovine and Goat Diseases, College of Animal Science and Technology, Guangxi Vocational University of Agriculture, Nanning 530007, China; 3Guangxi Zhuang Autonomous Region Engineering Research Center of Veterinary Biologics, Nanning 530004, China; 4Guangxi Key Laboratory of Animal Breeding, Disease Control and Prevention, Nanning 530004, China

**Keywords:** astroviruses, recombination, goat, new strains

## Abstract

Astroviruses are single-stranded, positive-sense RNA viruses capable of infecting humans as well as a wide range of mammalian and avian species, with a length of approximately 6.6–7.7 kb. In this study, 139 goat fecal samples collected from the Guangxi province were used for the RT-PCR detection, and two of these were positive for goat astrovirus, with a positivity rate of 1.44% (2/139). The complete genome sequence of an astrovirus strain and the partial genome sequence of a strain astrovirus, named GX WZ 2023 and GX HC 2023, were amplified and sequenced, and their sequence lengths were 6284 nt and 6213 nt, respectively. Among them, the capsid protein of goat astrovirus GX HC 2023 showed the highest amino acid identity of 95.9% with ovine astrovirus GX, which belonged to the MAstV-2 genotype. However, the closest relative of the GX WZ 2023 strain was found to be the caprine astrovirus Sichuan, with a nucleotide sequence identity of 76.8%. The ORF1ab nonstructural protein of this strain showed the highest amino acid identities of 89.2 and 95.8% with the ovine astrovirus S5.1 and caprine astrovirus G5.1 strains, respectively. However, its ORF2 capsid protein has 68.4% amino acid identity with the bovine astrovirus (BAstV) 16 2021 CHN strain and only 21.9–64% amino acid identity with all available strains of goat astrovirus. The GX WZ 2023 strain was recombined with the Chinese (BAstV 16 2021 CHN) and Japanese bovine strains (BAstV JPN 2015) in the ORF2 region. Therefore, the goat astrovirus GX WZ 2023 is proposed as a new member of the family goat astroviridae based on the species classification criteria of the International Committee on Taxonomy of Viruses. These findings enhance our understanding of the prevalence and genetic evolution of goat astrovirus and provide a scientific basis for future studies of these viruses in other animals.

## 1. Introduction

Astroviruses belong to the family Astroviridae, which is divided into two genera, Mamastrovirus (MAstV) and Avastrovirus, and these viruses infect mammalian and avian animals, respectively [1], with clinical symptoms including diarrhea, fever, vomiting, and encephalitis [2]. Astroviruses are envelope-less, single-stranded positive-sense RNA viruses, and they are 6–8 kb in length [1]. The genome is divided into three overlapping open reading frames, ORF1a, ORF1b, and ORF2, the first two of which are located at the 5′ end of the genome. Together, they encode the astroviral nonstructural proteins nsp1a and nsp1ab. The ORF2 is located at the 3′ end of the genome, and it encodes the astroviral capsid proteins [3]. It is divided into two regions, one of which is a highly conserved region near the N-terminus and another that is an extremely variable region near the C-terminus. It also has a cluster of viral antigenic determinants, which have multiple neutralizing epitopes [4,5]. The ORF1a front and the ORF2 tail contain the 5′UTR and 3′UTR non-coding regions, respectively, as well as a poly (A) tail at the 3′ end. In addition, there is a six-base pair stem and a loop region that have uncertain interactions between 31 nucleotides, which have been located near the ORF2 ending of the 3′ end of members of the family Astroviridae [6]. This genetic element corresponds to the second largest 3′ stem–loop structure (stem–loop II) in human astroviruses, and it was named s2m [7].

According to the ninth report of the International Committee for the Classification of Viruses (ICTV), the classification of MAstV is based on the analysis of the genetic distances of the intact capsid region at the amino acid level. However, within each genome and individual genotype of the species, there has to be at least 0.338 amino acid genetic distance of the capsid precursor protein [8]. Goat AstV was first discovered in Switzerland in 2019 by means of NGS technologies, and three genotypes (Caprine Astrovirus G5.1, Caprine Astrovirus G3.1, and MAstV–34) were discovered [9]. Other astrovirus genotypes that have been identified in goats or sheep include MAstV 13, MAstV 24 [10], MAstV-33 [11], and MAstV 2 [12]. The rich genetic diversity, the high frequency of recombination events, the ability to spread between different host species, and the recombination between different species, which contributes to cross-species transmission, are factors that have played important roles in the evolution of astroviruses [13].

In this study, two goat astrovirus strains were identified in Guangxi, China, and the genome sequences of these viruses were determined. Although astroviruses have been reported in goat and sheep fecal samples, the pathogenicity of potential goat astrovirus genotype species remains unknown due to the lack of cell culture systems and epidemiological survey data. Intraspecies and interspecies transmission and its relationship to disease are also poorly known. Therefore, there is a need to further investigate the relationship between astroviruses and goat diseases. The aim of this study was to investigate the diversity of goat astroviruses in Guangxi, and this study will help to understand the prevalence and evolution of goat astroviruses in Guangxi.

## 2. Materials and Methods

### 2.1. Sample Collection

A total of 139 goat fecal samples were collected from housed goats in Hechi (5 farms) and Wuzhou (2 farms) of Guangxi province, China. None of the goats had any obvious diarrhea symptoms, and their ages ranged from 20 days to 6 months. Autoclaved 10 mL centrifuge tubes were used to collect the samples in volumes ranging from 3 to 5 mL per sample. All samples were transported on ice and stored at −80 °C.

### 2.2. RNA Extraction

The fecal samples were diluted to produce 10% suspensions with PBS, and these were centrifuged at 4 °C at 3000 r/min for 10 min. Viral RNA was extracted from the supernatants by using a Viral DNA/RNA Mini Prep kit (Axygen, Hangzhou, China) according to the manufacturer’s instructions.

### 2.3. Goat Astrovirus Detection

Goat astrovirus was detected based on our previous description [11]. The forward and reverse primer sequences were AstV-F, 5′-CTTTGGAGGGGGMGGGACCAA-3′, and AstV-R, 5′-TCTGGAAAACCACACACGGT-3′, respectively. Viral RNA was reverse transcribed using the downstream primers to obtain the cDNA. PCR amplification was performed in a 25 μL reaction volume of 0.5 μlL containing 10 μM forward primer, 10 μM reverse primer, 3 μL cDNA, 12.5 μL 2 × Rapid Taq Master Mix (Vazyme, Nanjing, China), and the appropriate amount of double-distilled water. Pre-denaturation was at 95 °C for 5 min, followed by 35 cycles of denaturation at 95 °C for 15 s, annealing at 54 °C for 15 s, extension at 72 °C for 10 s, and extension at 72 °C for 5~10 min. At the end of the reaction, the PCR products were separated by 1.5% agarose gel electrophoresis and viewed by ultraviolet imaging. The identified products were sequenced by the Guangzhou Huada Gene Sequencing Company.

### 2.4. Complete Genome Sequencing and Sequence Analysis

To amplify the genome sequences of goat astrovirus, specific primers (Appendix A) were designed according to the reference sequence of goat astrovirus available from the NCBI. The target sequences were amplified by RT-PCR, as described above. The PCR products were purified by using a Tiangen’s DNA Universal Product Purification and Recovery Kit (Tiangen, Beijing, China) and subsequently cloned into the pMD™18-T vector (Takara, Dalian, China). Sequence splicing was performed using the SeqMan application in the DNASTAR 7.0 software package (DNASTAR Inc., Madison, WI, USA), and the MegAlign application was used to determine the nucleotide and amino acid sequence identities. The online site UNAFold (http://www.unafold.org/, accessed on 1 March 2024) was utilized to predict whether the virus had a s2m structure.

### 2.5. Phylogenetic and Recombination Analyses

Phylogenetic trees based on the full-length genome, ORF1a, ORF1b, and ORF2 sequences were constructed using the neighbor-joining method in MEGA 7.0 software. The genetic evolutionary distances were calculated using *p*-distance, and the internal nodes were evaluated by performing 1000 repetitions of the self-test. Recombination analysis was performed in RDP 4.0 software using the default parameter values set to analyze possible recombinant strains as well as their parental sequences. Sequence analysis of the nucleotides at recombinant positions was compared using the ClustalW method in MEGA 7.0 software.

## 3. Results

### 3.1. Detection of Goat Astrovirus

A total of 139 goat fecal samples were collected from 7 farms in Guangxi, China. The goat astroviruses were detected by RT-PCR. Two of the fecal samples were positive for goat astrovirus, with a positivity rate of 1.44% (2/139). The two strains detected in this study were named goat astrovirus GX HC 2023 and goat astrovirus GX WZ 2023, respectively. The sequences of the GX HC 2023 and GX WZ 2023 were submitted to GenBank under the accession numbers PP571882 and PP646881, respectively.

### 3.2. Genomic Characterization

The sequence lengths of GX HC 2023 and GX WZ 2023 were 6213 nt and 6284 nt, respectively, and the GC content was 49% for both. The nucleotide sequences of the genomes and the amino acid sequences of the ORFs were deduced after comparing the sequences with those of other astroviruses (Figure 1). The genomes of the two strains were found to have a ribosomal shift site (5′-AAAAAAC-3′) at the 3′ end of ORF1a at the overlap of ORF1a and ORF1b. This sequence is essential for the translation of RNA-dependent RNA polymerase (RdRp), which is encoded in ORF1b, and it was found to be present exactly in this overlapping region [14,15].

#### 3.2.1. Goat Astrovirus GX HC 2023 Genomic Characterization

The genome of goat astrovirus GX HC 2023 contains 3′UTR, which is 67 nt; 5′UTR is uncertain. The genome also contains a partial ORF1a sequence. ORF1a is 2432 nt in length and encodes 809 amino acids, forming the nonstructural protein 1a. ORF1b is 1508 nt in length and encodes 501 amino acids. There is a 42 nt overlap between ORF1a and ORF1b. ORF2 is 2259 nt in length and encodes 752 amino acids, which form the coat protein precursor. The identity of the nucleotide and deduced amino acid sequences of goat astrovirus GX HC 2023 and goat astrovirus GX WZ 2023 were compared with other astrovirus strains (Appendix A). The goat astrovirus GX HC 2023 was found to share 94.3% nucleotide identity with the ovine astrovirus GX strain at the genome level. Its ORF1a and ORF1b showed the highest nucleotide and amino acid identities (95.1–98.1%) with the ovine astrovirus GX as well as the caprine astrovirus SWUN F1 2019 (GenBank accession no. OK107512) strains. Similarly, ORF2 showed the highest nucleotide and amino acid identities with the ovine astrovirus GX strain at 93.2% and 95.9%, respectively.

The phylogenetic evolutionary trees were constructed based on nucleotide sequences of the genome of the virus we obtained for the goat astrovirus GX HC 2023 and goat astrovirus GX WZ 2023 strains, the amino acid sequences of ORF1a, ORF1b, and ORF2, and the reference sequences of other astroviruses from the GenBank (Figure 2a–d). The phylogenetic tree analysis revealed that goat astrovirus GX HC 2023 clustered with the same branch as ovine astrovirus GX with respect to the genome as well as with the amino acid sequences of ORF1a, ORF1b, and ORF2, respectively. In addition, they belonged to the MAstV-2 subtype.

#### 3.2.2. Goat Astrovirus GX WZ 2023 Genomic Characterization

The genome of goat astrovirus GX WZ 2023 contains a 5′UTR and a 3′UTR, which are 23 nt and 192 nt, respectively. ORF1a is 2442 nt in length and encodes 813 amino acids, forming the nonstructural protein 1a. ORF1b is 1507 nt in length and encodes 501 amino acids. There is a 44 nt overlap between ORF1a and ORF1b. ORF2 is 2172 nt in length and encodes 723 amino acids. Similarly, by using BLAST analysis, we found that the goat astrovirus GX WZ 2023 strain is closely related to the caprine astrovirus Sichuan (GenBank accession no. MW784100) at the whole genome level, with a nucleotide identity of 76.8%. In addition, the nucleotide and amino acid identities of ORF1a and ORF1b of this strain with other goat astrovirus strains ranged from 38.4 to 90.6% and 24.5 to 95.4%, respectively. In addition, the ORF2 nucleotide identity of this strain was the highest with caprine astrovirus Sichuan and bovine astrovirus (BAstV) 16 2021 CHN (GenBank accession no. ON624266) strains at 69.8 and 68.9%, respectively, while the amino acid identity was highest with BAstV 16 2021 CHN at 68.4%.

We subsequently compared the distances of coat proteins between the goat astrovirus GX WZ 2023 strain and all available goat and sheep astroviruses (*p* distance > 0.338; Table 1). Phylogenetic trees were constructed by the maximum likelihood method, and the analysis showed that the whole genome of the goat astrovirus GX WZ 2023 strain and the amino acids from ORF1a and ORF1b clustered in the same branch as the other goat astrovirus strains. However, the phylogenetic evolutionary tree constructed based on the ORF2 amino acids indicated that the goat astrovirus GX WZ 2023 strain was similar to the Chinese BAstV 16 2021 CHN obtained from cattle, and they clustered on a small independent branch, which, in turn, clustered on a large independent branch with caprine astrovirus Sichuan (Figure 2d).

#### 3.2.3. s2m Stem–Loop Structure

When the ORF2 ending in the 3′UTR nucleotide sequences of goat astroviruses and other mammalian astroviruses currently available at NCBI were compared against the goat astrovirus GX HC 2023 and goat astrovirus GX WZ 2023 strains, it was found that the two strains isolated in this study did not have an s2m stem–loop structures. Equally surprising, there are also goat astroviruses that do not have a stem–loop structure. In addition, the sequences of the goat astroviruses that did not have an s2m stem–loop structure had deletions at the same positions and some nucleotide sequence similarities (Figure 3). However, for the astroviruses that did have s2m stem–loop structures, they consisted of 43 nt and were basically identical to those found in other astroviruses. The structures were located at the 24th nucleotide, from the end of the ORF2 to the end of the 3′UTR. 

### 3.3. Recombination Analysis

Recombination analysis was performed on the whole genome sequences of known goat astroviruses and other species of astroviruses, with the recombination events predicted to occur only in goat astrovirus GX HC 2023 and goat astrovirus GX WZ 2023. The genome sequences of all the above astroviruses were compared using the Clustal W method in MEGA 7.0 software, and the final output files were in Fasta format. The aligned fasta files were imported into RDP4 software (RDP 4.0, version 4.96) for recombination analysis. Purification analysis by RDP, GeneConv, Chimaera, MaxChi, BootScan, SiScan, and 3Seq methods using the RDP4 revealed two sets of recombinants, then the overview toolbar displays the analysis results and outputs the image results in RDP format. These were found in different regions of the ORF2 (Figure 4a,b).

The caprine astrovirus Sichuan strain was recombinant with ovine astrovirus S5.1 (GenBank accession no. MK404648) as the primary parent and goat astrovirus GX WZ 2023 as the secondary parent, with recombination positions at nucleotides 1782 and 5502 (Figure 4a). In addition, in another set of recombination events, we found the BAstV 16 2021 CHN strain to be recombinant, with the putative primary and secondary parents being BAstV JPN 2015 (GenBank accession no. LC047797) and goat astrovirus GX WZ 2023 strains, with the breakpoints located at nt positions 4584 and 5808, respectively (Figure 4b). It was found that all of the above recombination events occurred within ORF2. A comparison of the nucleotide sequences at the ORF2 position of the recombinant strains in which they occurred was carried out using Clustal W in the MEGA 7.0 software. This revealed that goat astrovirus GX WZ 2023 was essentially similar to the BAstV 16 2021 CHN strain when the ORF2 nucleotides at positions 326–359, 391–401, and 406–425 were compared (Figure 5). Here, we were able to show how the goat astrovirus GX WZ 2023 strain and BAstV underwent recombination (Figure 6). Under natural conditions, astroviruses can infect goats and bovines due to their wide host range; these two viruses then underwent a series of recombination reactions in goats to form the goat astrovirus GX WZ 2023 strain. 

## 4. Discussion

This study describes the discovery and molecular characterization of two astrovirus genotypes in goats, which show extensive genetic diversity when compared to AstV in other species. Sheep astrovirus was first identified in animals in 1977 in sheep fecal samples [16], but its biological significance in sheep is unclear. There is little known regarding sheep astrovirus infection in ruminants as well as its intra-species and inter-species transmission and its relationship to disease. Here, 139 goat fecal samples from Hechi City (5 farms) and Wuzhou City (2 farms) in Guangxi, China, were assessed for the presence of astroviruses, and a positivity rate of 1.44% was found. This indicated that goat astroviruses were endemic in Guangxi, China. In another study using 143 fecal samples collected from 11 goat farms in Southwest China that were tested for goat astrovirus by RT-PCR, a positive rate of 3.7% was found [11]. Some authors have reported that asymptomatic infections are relatively common with respect to mammalian astroviruses [17,18,19]. However, as the feces collected in this study were from animals without diarrhea symptoms, it was not possible to determine whether there was any association between goat astrovirus infection and diarrhea.

It was shown that although the s2m stem–loop structure is an essential element in the life cycle of mammalian viruses [7], in this study, after predicting and comparing with other goat astroviruses, in both the novel strains of goat astroviruses, there were no s2m stem–loop structures, indicating that not all goat astroviruses contain these. Since the biological role of the s2m motifs is still unclear, we hypothesize that these strains are able to form stable secondary structures without s2m motifs. However, whether or not these viruses have compensatory elements that have a similar role to that of the s2m motif needs to be further investigated.

The results of recombination analysis were generally consistent with the phylogenetic evolutionary tree and amino acid identity analysis of individual genomes. In addition, their classification as MAstVs depended on the genetic distance between the original host and the full-length coat protein according to the species classification criteria of the ICTV (>0.338). In this study, a phylogenetic evolutionary tree was constructed based on the ORF2 amino acid sequence, and it was found that goat astrovirus GX HC 2023 clustered in the same branch with ovine astrovirus GX, and the nucleotide and amino acid identities were 93.2 and 95.9%, respectively. Since ovine astrovirus GX belongs to the MAstV-2 subtype, goat astrovirus GX HC 2023 is considered a new member of this group of viruses. The ORF2 of the goat astrovirus GX WZ 2023 strain has 21.6–64% amino acid identity with other sheep astroviruses, and it has the highest amino acid identity (68.4%) with BAstV 16 2021 CHN. It is also in the same branch of the ORF2 amino acid evolutionary tree, and when compared to the coat protein genetic distances of all sheep astroviruses, they were all found to be greater than 0.338. This suggests that the goat astrovirus GX WZ 2023 strain may represent a new genotype species in goats.

In recent years, mutational events either between or within astrovirus genotypes have been increasingly reported as a major cause for the emergence of new astrovirus universal lineages [20,21]. Recombination events have also been frequently reported to exist between Sika deer astroviruses and BAstVs [22], between porcine astroviruses and HAstV-3 [21], and between sheep astroviruses and goats [9]. BoAstV-CH 15 and OvAstV-CH 16 were observed in astrovirus-associated encephalitis in a previous ground study [23], and the prediction of recombination events further justifies the existence of interspecies transmission events since recombination between viruses requires co-infection in the same host cells [22]. In this study, we unexpectedly found that goat astrovirus GX WZ 2023 recombined with ovine astrovirus S5.1 and caprine astrovirus Sichuan strains in the ORF2 region. Goat astrovirus (goat astrovirus GX WZ 2023) and BAstV (BAstV 16 2021 CHN and BAstV JPN 2015) also showed recombination events in the ORF2 region, and the nucleotide sequences in the recombinant regions showed a certain degree of identity. Recombination in different regions of ORF2 may diversify the viral capsid proteins, which may affect the antigenicity of the viruses, allowing them to evade host immunity [1,24]. In addition, these events can contribute to the genetic diversity of the astrovirus population.

In a previous study from our laboratory, we confirmed that when 532 healthy pig fecal samples were collected from 2013 to 2015 in Guangxi Province, China, and tested for porcine astrovirus, there was a positivity rate of 56.4% [25]. Another study tested 211 fecal samples collected from calves with mild to severe diarrhea by RT-PCR, and the BAstV positivity rate was 43.6% [26]. Although our study suggests that the overall prevalence of astroviruses in goats is low compared to that in other hosts, transmission of these viruses between different animals is possible. 

This study identified, for the first time, recombination events of astroviruses within a goat population as well as between goats and cattle in Guangxi. In addition, we identified a new genotype in the goat genome, which contributes to a better understanding of the genetic diversity and evolution of astroviruses in goats.

## Figures and Tables

**Figure 1 viruses-16-01213-f001:**
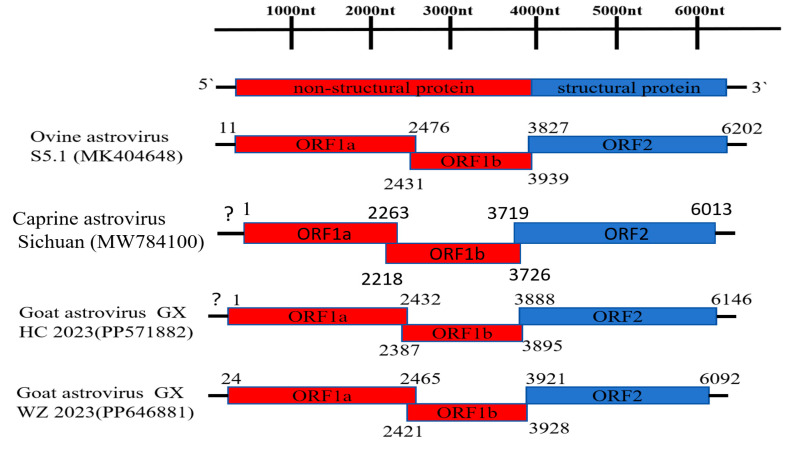
Schematic representation of the genomes of ovine astrovirus S5.1 and caprine astrovirus Sichuan as a reference strain with goat astrovirus GX HC 2023 and goat astrovirus GX WZ 2023. Their genomes consist of 5′ and 3′ untranslated regions (UTRs) at both ends and a middle region, which encodes the nonstructural proteins ORF1a and ORF1b and the coat protein (ORF2). “?” Indicates that the sequence information of the 5′ UTR is uncertain.

**Figure 2 viruses-16-01213-f002:**
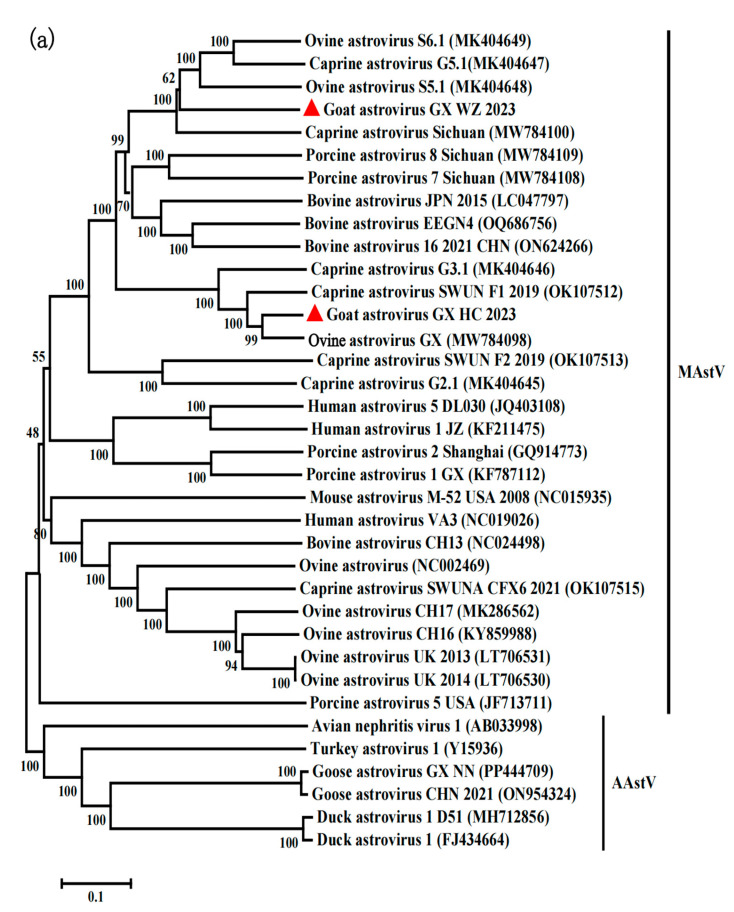
Phylogenetic tree based on the complete amino acid sequences of the viral genome sequence and nucleotide sequences (**a**), ORF1a (**b**), ORF1b (**c**), and ORF2 (**d**) using ClustalW in the MEGA 7.0 software for sequence comparison and clustering. Phylogenetic evolutionary trees were constructed using the neighbor-joining method, and bootstrap values were calculated for 1000 replicates. The two new goat astrovirus strains isolated in this study are labeled with red triangles.

**Figure 3 viruses-16-01213-f003:**
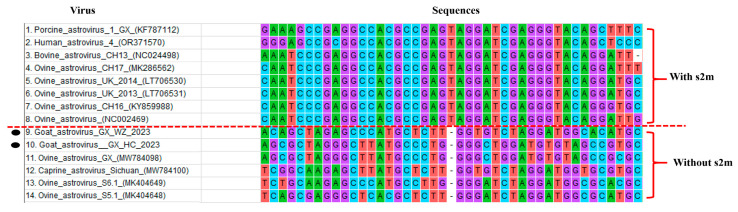
Nucleotide sequences of the s2m stem–loop structures obtained based on the predicted ORF2-terminal to the 3′UTR multiple sequence comparison between the genomes of goat astrovirus and other mammalian astroviruses available at NCBI and the novel goat astrovirus GX HC 2023 and GX WZ 2023 strains in this study. The sequences above and below the red dashed line represent the strains with and without s2m stem–loop structures, respectively. The two novel strains in this study are indicated by black circles.

**Figure 4 viruses-16-01213-f004:**
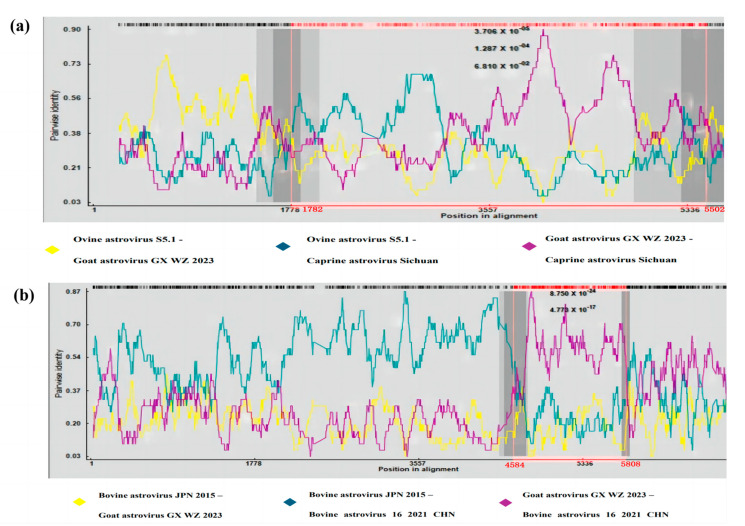
Recombination analysis of the genome of the goat astrovirus GX WZ 2023 strain with the predicted recombination events (**a**,**b**). (**a**) Predicted recombination event 1 with caprine astrovirus Sichuan as the recombinant, ovine astrovirus S5.1 as the primary parent, and goat astrovirus GX WZ 2023 as the secondary parent. (**b**) Predicted recombination event 2, in which bovine astrovirus 16 2021 CHN serves as the recombinant, bovine astrovirus JPN 2015 serves as the primary parent, and goat astrovirus GX WZ 2023 serves as the secondary parent. The constructs were graphically depicted by using the RDP method in RDP 4. Nucleotide positions of astrovirus are depicted on the axis of abscissas in kb. The red bars schematically mark the portion of the genome involved in the recombination event.

**Figure 5 viruses-16-01213-f005:**
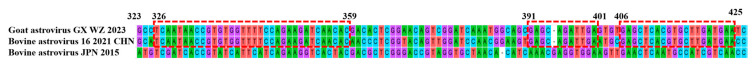
Sequence comparison of nucleotides at the partial ORF2 recombination positions against strains of goat astroviruses that have recombined with bovine astrovirus. Goat astrovirus GX WZ 2023 and bovine astrovirus 16 2021 CHN contain the same regions, which are marked with a red box. The nucleotide positions are based on the goat astrovirus GX WZ 2023 strain.

**Figure 6 viruses-16-01213-f006:**
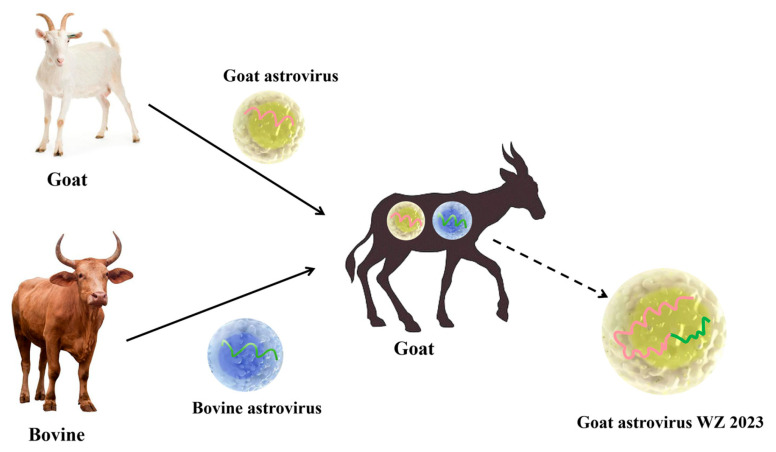
Schematic diagram of goat astrovirus recombining with bovine astrovirus to form the goat astrovirus GX WZ 2023 strain.

**Table 1 viruses-16-01213-t001:** *p*-distances for the goat astrovirus GX WZ 2023 strain and all available sheep and goat astrovirus capsid proteins.

Strain	GenBank Accession No.	*p*-Distance
Goat astrovirus GX HC 2023	PP571882	0.556
Caprine astrovirus SWUN F2 2019	OK107513	0.662
Caprine astrovirus SWUN F1 2019	OK107512	0.568
Caprine astrovirus Sichuan	MW784100	0.359
Ovine astrovirus GX	MW784098	0.557
Caprine astrovirus G5.1	MK404647	0.456
Caprine astrovirus G3.1	MK404646	0.558
Caprine astrovirus G2.1	MK404645	0.672
Ovine astrovirus UK 2014	LT706530	0.789
Ovine astrovirus UK 2013	LT706531	0.789
Ovine astrovirus S6.1	MK404649	0.463
Ovine astrovirus S5.1	MK404648	0.44
Ovine astrovirus CH17	MK286562	0.789
Ovine astrovirus CH16	KY859988	0.792
Ovine astrovirus	NC002469	0.799

## Data Availability

The sequences of goat astrovirus GX HC 2023 and goat astrovirus GX WZ 2023 described in this study have been deposited in GenBank under accession numbers PP571882 and PP646881, respectively.

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
