# Peer review of "Identification and Full-Length Sequence Analysis of a Novel Recombinant Goat Astrovirus Genotype in Guangxi, China"

_viruses, 2024, doi:10.3390/v16081213_

Round 1
Reviewer 1 Report
Comments and Suggestions for Authors
The study of Yulu Zhou and co-workers describes the RT-PCR-based detection and phylogenetic/sequence analyses of two (strain GX WZ 2023 and GX HC 2023) novel astroviruses from goats. The GX HC 2023 strain most likely belongs to a same genotype species (MAstV-2) as its closest relative (caprine AstV GX) while the GX WZ 2023 is most likely a founding member of a novel genotype species due to the low capsid amino acid sequence identity (p-dist >0.338) to any of the known AstVs. The authors also stated that GX WZ 2023 is a recombinant strain.
The findings are somewhat interesting, but the manuscript is too long and did not contain enough results to justify the format (research article), it should be rewrite as a short communication, furthermore the study contains multiple flaws and scientific errors, and misleading analyses (see the detailed points below).
Major points
One of the two main problems with the study is (i) that there is no convincing evidence which support the recombination nature of GX WZ 2023 because the most closely related viral sequences in the ORF2 shows only 69.8% (Caprine AstV Sichuan) and 68.9% (Bovine AstV 16 2021 CHN) nt identity to GX WZ 2023 (Table 4) which means that none of them could serve as potential donors for a recombination of which results the emergence of GX WZ 2023. The real parental sequences are missing. Therefore, the sections (lines 235-251) about the recombination analysis should be rephrased more cautiously. E.g. GX WZ 2023 could be a recombinant strain, but the sequence similarity between the presumed parental sequences could be the results of adaptive evolutional changes.
(ii) The second main problem is that the authors state that “The full-length sequences of GX HC 2023 and GX WZ 2023 were 6213 nt and 6284 nt, respectively” (in lines 125-126), but according to the GenBank flatfile of GX HC 2023 (PP571882), although the sequence is indeed 6213-nt-long, but far from complete, it’s 5’ end is missing, including the 5’ UTR and the start of ORF1a. I don’t know where the authors got the information about the organization of the 5’ end of the genome of GX HC 2023 in figure 1? Furthermore, the MW784098 is an ovine AstV not caprine.
Details of cloning and sequencing of the PCR products as well as the methods used for the determination of the 5’ and 3’ ends are completely missing. How many clones did you sequenced? Why it was necessary to clone the PCR products?
Details of the recombination analysis is completely missing
Minor points
Lines 20, 23, 183, 185, 285…: The authors are incorrectly using the term “homology” throughout the entire manuscript. I think the term “identity” is more appropriate in case of sequence comparisons.
Lines 95, 147, 180, 234, The term “gene” should be avoided. There is no “gene” in the genome of AstV only genome parts/regions.
Table 1 should contain more information. e.g. primer binding sites in a reference genome. And should be moved to supplementary.
Figure 4: The resolution of the image is too low, and this is not a recombination analysis but an identity plot. It shows that different parts of the genome show different level of identity to the sequences used for the analysis.
All the tables should be supplemented with accession numbers of the strains, and all should be moved to supplementary.
There are lots of inappropriate/non-scientific phrases which should be re-phrased. E.g. “whole gene level” (lines 147, 180), “viral genome wide nucleotides” (lines 156-157), “complete 6069 bp ORF” (line 174), “virulent strain” (line 227), “intraspecific and interspecific” (line 265).
Author Response
Response to Reviewer’s Comment 1: Manuscript ID:viruses-3056036
Original Manuscript Title: Identification and Full-Length Sequence
Analysis of a Novel Recombinant Goat Astrovirus Genotype in Guangxi, China
We faithfully thank the editors and all reviewers for their valuable suggestions! As you are concerned, there are several issues that need to be addressed. Based on your suggestions, we have revised and corrected our previous manuscript with the following corrections:
Specific Comments
Reviewer #1
- that there is no convincing evidence which support the recombination nature of GX WZ 2023 because the most closely related viral sequences in the ORF2 shows only 69.8% (Caprine AstV Sichuan) and 68.9% (Bovine AstV 16 2021 CHN) nt identity to GX WZ 2023 (Table 4) which means that none of them could serve as potential donors for a recombination of which results the emergence of GX WZ 2023. The real parental sequences are missing. Therefore, the sections (lines235-251)about the recombination analysis should be rephrased more cautiously. E.g. GX WZ 2023 could be a recombinant strain, but the sequence similarity between the presumed parental sequences could be the results of adaptive evolutional changes.
Response: Thank you very much for your careful reading, we think this is a great suggestion and we have rewritten this section based on your suggestions. And it is marked in red font in the text. see lines 245-252.
- The second main problem is that the authors state that “The full-length sequences of GX HC 2023 and GX WZ 2023 were 6213 nt and 6284 nt, respectively” (in lines 125-126), but according to the GenBank flatfile of GX HC 2023 (PP571882), although the sequence is indeed 6213-nt-long, but far from complete, it’s 5’ end is missing, including the 5’ UTR and the start of ORF1a. I don’t know where the authors got the information about the organization of the 5’ end of the genome of GX HC 2023 in figure 1?
Response: Thank you to the reviewers for their suggestions. For information about the sequence of the 5' end of the GX HC 2023 genome, I designed primers for amplification of the 5' end based on the 5' end of the reference sequence caprine astrovirus GX (MW784098).
- Details of cloning and sequencing of the PCR products as well as the methods used for the determination of the 5’ and 3’ ends are completely missing. How many clones did you sequenced? Why it was necessary to clone the PCR products?
Response:Thank you for pointing this out. Regarding how to determine the 5' end and 3' end in the article the method used was based on the ovine astrovirus available on NCBI as a reference sequence, primers were designed to amplify the 5' end and 3' end of the reference sequence, and considering that the amplified fragments were not very long, each amplified fragment was cloned, and the cloning was done because some of the sequences at the beginning and end of the sequencing reaction were not accurate enough, so it would be best to clone the sequences to be sequenced into the vector for sequencing.
- Details of the recombination analysis is completely missing.
Response: We thank the reviewers for their suggestions. We have re-improved this section, and the details of the reorganization analysis can be found in 3.3. Reorganization Analysis, highlighted in red font in the article. See lines 219-226.
- Lines 20, 23, 183, 185, 285…: The authors are incorrectly using the term “homology” throughout the entire manuscript. I think the term “identity” is more appropriate in case of sequence comparisons.
Response: Thank you very much for your suggestion, and I apologize for not noticing it at first. Based on your comment, we have made a correction to make the word consistent throughout the manuscript, and all references to “homology” in the article have been changed to “identity” and highlighted in red.
- Lines 95, 147, 180, 234, The term “gene” should be avoided. There is no “gene” in the genome of AstV only genome parts/regions.
Response:Thank you for pointing this out. I agree with this comment. we have made the following changes to our previous manuscripts.
Lines 95, The word “gene” has been corrected and replaced with the word “gene”or “sequences”. See lines 102.
Line 147, 180, 234, The word “gene” has been corrected and replaced with the word “gene”or “genome”, And all marked in red in the article. See lines 150,179,235,
- Table 1 should contain more information. e.g. primer binding sites in a reference genome. And should be moved to supplementary.
Response:We think this is a good suggestion, and we have supplemented Tables1 and 2 with the primer binding sites in the reference genome, and should the tables be moved to the Supplementary section now.
- Figure 4: The resolution of the image is too low, and this is not a recombination analysis but an identity plot. It shows that different parts of the genome show different level of identity to the sequences used for the analysis.
Response: We apologize for our carelessness and thank you for your suggestion, we have corrected it by increasing the resolution of all the images in the article, more than just Figure 4.
- All the tables should be supplemented with accession numbers of the strains, and all should be moved to supplementary.
Response: Thank you very much for your suggestion, indeed we have considered this problem before, but due to the long naming of many strains in the table, the whole table is too big and not beautiful enough after adding the strain's registration number column, which reduces the comfort level of the readers. In order to solve this problem, all the strains used in the table of the article are the same as the serial numbers of the strains used in the evolutionary tree. This makes it easy for the reader to see the registration number of each strain, and tables has been moved to the supplementary section.
- There are lots of inappropriate/non-scientific phrases which should be re-phrased. E.g. “whole gene level” (lines 147, 180), “viral genome wide nucleotides” (lines 156-157), “complete 6069 bp ORF” (line 174), “virulent strain” (line 227), “intraspecific and interspecific” (line 265).
Response:Thank you for your careful scrutiny, we apologize for our careless mistakes, and in response to your reminder, we have corrected all of them, with the corrections marked in red in the text, see lines 150, 179, 155-156, 173-174, 228, 266. We hope the revisions will be recognized.

Reviewer 2 Report
Comments and Suggestions for Authors
Dear Authors
The manuscript describes the discovery of a novel strain of astrovirus in a goat that was confirmed to be a recombinant with bovine astroviruses. The work was neatly structured and easy to follow.
Please consider the following recommendations:
1. Introduction
1.1 The aim of the study is not clearly stated at the end of the introduction (Page 2, lines 65 to 67).
1.2 Page 2, line 57: Add ref 19 to " Goat astroviruses were first discovered in Switzerland..."
2. Material and Methods
2.1 Page 3, Tables 1 and 2: Consider "Primer pair sequence" for table column headings, since primer pairs are given
2.2 Tables should not be presented over two pages, revise table size
2.3 Page 4, line 112 states recombinant analyses was performed using RDP 4.0 software. More details of the programs used is needed in order to understand results mentioned in results-page 11, line 222
3. Results
3.1 Results from RDP 4.0 analyses should be given to the reader. This may be a table in text or as supplementary information. Such information is important for readers who want to reproduce the same work.
3.2 Tables 3, 4 and 5: adjust table size to fit on one continuous page
3.3 Figure 2: Increase size of image. Text in image is not sharp and makes reading difficult. Bootstrap values and scale bar - text is too small
3.4 Figure 5: Increase size of image. Text in image is too small to see easily
3.5 Page 12, line 227 (Figure 4 legend): strain GX WZ 2023 is referred to as "virulent." However, in sample collection (section 2.1, page 2, line 71) it is stated "none of the goats had any diarrhea symptoms." How is virulence of GX WZ 2023 determined?
3.6 Page 12, line 238: see sentence structure
3.7 Page 12, line 249: please revise this sentence or provide a reference for "it is known that goat astroviruses and BAstVs can co-infect goats."
4. Discussion
4.1 Page 13, line 276: Consider starting sentence with "although." In this study the stem-loop structure was not detected, thus the statement in line 276, "It has been shown.......is an essential element in the life cycle...." is contradicted by the result obtained. The hypothesis offered in line 280 stands and is sufficient.
4.2 Page 14, line 322: these study conclusions are important. Consider presenting them in their own paragraph for more impact.
Comments on the Quality of English LanguageMinor language and text edits are required:
1. Page 2, line 57: add a space between "[8]." and "Goat"
2. Page 2, line 83: add word "and" between forward and reverse sequences
3. Page 11, line 204: revise sentence structure, "that there were other some of the goat astroviruses"
4. Page 12, line 238: revise sentence structure, "we it was found "
Author Response
Response to Reviewer’s Comment: Manuscript ID:viruses-3056036
Original Manuscript Title: Identification and Full-Length Sequence
Analysis of a Novel Recombinant Goat Astrovirus Genotype in Guangxi, China
Dear reviewers, We appreciate your professional comments on our article. As you are concerned, there are several issues that need to be addressed. Based on your suggestions, we have revised and corrected our previous manuscript with the following corrections:
1.The aim of the study is not clearly stated at the end of the introduction (Page 2, lines 65 to 67).
Response: Thank you very much for your careful reading, we apologize for our carelessness, and based on your suggestion, we have made a correction by adding the purpose of the article's research, in lines 65-73 of the article and marking it in red.
2.Page 2, line 57: Add ref 19 to " Goat astroviruses were first discovered in Switzerland..."
Response: Thank you very much for your advice and I apologize for our carelessness. I made a mistake in the original manuscript, and we have corrected it in the article by adding references, which are marked in red in lines 57-60 of the article.
3.Page 3, Tables 1 and 2: Consider "Primer pair sequence" for table column headings, since primer pairs are given.
Response: Thank you for pointing this out. I agree with this comment.
Thank you for your suggestion, we have used “Primer Pair Sequences” as the tabular title and moved the table to the Supplementary section, where it can be found and labeled in red font.
4.Tables should not be presented over two pages, revise table size.
Response: Thank you very much for your comments, we agree and have made corrections, resized the table and have moved it to the supplemental section based on reviewer comments.
- Page 4, line 112 states recombinant analyses was performed using RDP 4.0 software. More details of the programs used is needed in order to understand results mentioned in results-page 11, line 222.
Response: Thank you very much for your suggestion. We have repopulated the article with detailed steps for using RDP4, highlighted in red font. See line 219-226. We hope the revised manuscript is acceptable to you!
- Results from RDP 4.0 analyses should be given to the reader. This may be a table in text or as supplementary information. Such information is important for readers who want to reproduce the same work.
Response: Thank you for your suggestion. Regarding the results of the RDP 4.0 analysis, we have also referred to other articles on how to present the results of the reorganization analysis, so the results of the reorganization analysis in this article are presented as a note in Figure 4, See line 228-235.
7.Tables 3, 4 and 5: adjust table size to fit on one continuous page.
Response:We think this is a good suggestion, and we have all but resized the tables so that they fit on one continuous page and can be found in the Supplementary section based on other reviewer suggestions that Tables 3 and 4 have been moved to the Supplementary section.
- Figure 2: Increase size of image. Text in image is not sharp and makes reading difficult. Bootstrap values and scale bar - text is too small.
Response:Sincerely thank you for your valuable suggestions, we have corrected the picture, we hope the revised picture can make you satisfied.
- Figure 5: Increase size of image. Text in image is too small to see easily
Response:Sincerely thank you for your valuable suggestions, we have re-increased the image size and the text size in the image to give a comfortable feeling to the readers.
- Page 12, line 227 (Figure 4 legend): strain GX WZ 2023 is referred to as "virulent." However, in sample collection (section 2.1, page 2, line 71) it is stated "none of the goats had any diarrhea symptoms." How is virulence of GX WZ 2023 determined?
Response:Thank you very much for your careful scrutiny and we apologize for our carelessness. Although several attempts were made to isolate the strain WZ 2023 in this article by cell isolation and culture techniques, it was not successfully isolated, so its virulence could not be determined. Therefore, “virulent” in the original article was a misrepresentation on my part, and I deeply apologize for that, we have redone it and marked it in red font in the article, see line 228.
- Page 12, line 238: see sentence structure
Response:Thank you very much for the heads up, we have re-structured the sentence and marked it in red in the text, See line 239-242.
- Page 12, line 249: please revise this sentence or provide a reference for "it is known that goat astroviruses and BAstVs can co-infect goats."
Response:Thank you very much for your suggestion, we have carefully examined the manuscript and found that it was indeed a misrepresentation on my part, we have revised the manuscript and the revised document is marked in red in the text. see line 248-249.
- Page 13, line 276: Consider starting sentence with "although." In this study the stem-loop structure was not detected, thus the statement in line 276, "It has been shown.......is an essential element in the life cycle...." is contradicted by the result obtained. The hypothesis offered in line 280 stands and is sufficient.
Response:Thank you very much for the valuable advice you have provided, we couldn't agree more with your suggestion and we have re-corrected the sentence, marking it in red font in the article, see line 277.
14.Page 14, line 322: these study conclusions are important.Consider presenting them in their own paragraph for more impact.
Response:Thank you for your suggestion, we have re-corrected it and put it in its own paragraph. Mark it in red font in the article.
- Page 2, line 57: add a space between "[8]." and "Goat"
Response:We apologize for our carelessness, and thanks to your reminder, we have redone the correction, add a space between “[8].” and “Goat” and highlighted it in red font in the article. see line 57.
16.Page 2, line 83: add word "and" between forward and reverse sequences.
Response:Thank you for your suggestion, we have redone the correction and highlighted it in red in the text. See line 88.
17.Page 11, line 204: revise sentence structure, "that there were other some of the goat astroviruses"
响应:感谢您的建议,我们重新设计了句子结构,并在文本中以红色突出显示。请参阅第 202-203 行。
- 第 12 页,第 238 行:修改句子结构,“我们被发现”
响应:感谢您的提醒,感谢您的建议,这与问题 11 是同一个问题,我们重新设计了文章的句子结构,希望修改后的稿件能被您接受。

Reviewer 3 Report
Comments and Suggestions for Authors
This article describes the detection of two strains of goat astrovirus from goat farms in Guangxi Province, and analyzes in detail the genomic characterization of the two strains, one of which was identified as a new genotype of goat astrovirus with recombination with bovine astrovirus, with interesting and complete results. But there are some minor problems with the article:1.The picture in Figure 2 is not clear, please adjust the picture sharpness.2.Line 86, for "2 × Rapid taq Master Mix", please provide the name of the reagent supplier.3.The paper mentions that different primers were used to amplify the viral genome, but the annealing temperatures used are not seen in Tables 1 and 2.so please add the annealing temperatures of the primers used.4.Line 158, "goat astrovirus GX WZ" should be "goat astrovirus GX WZ 2023", please use the same expression throughout the text.5. Hyperlinks should be added to sequences mentioned in the article to facilitate quick access to sequence information by the reader. For example, line 149, lines 180-185, 236-240.6. The evolutionary tree constructed in the article used fewer reference sequences for ovine, which could be increased.7.Lines 43, 158 and 224, "ORF 2" is incorrectly formatted, please check the whole text and correct it uniformly.8.Lines 212 and 216, "s2 m" is incorrectly formatted, please check the whole text and correct it consistently.9.In line 263, the Astv in "AstV was first identified in animals in 1977 in sheep fecal samples" is incorrectly stated, confirming whether it is Astv or sheep astrovirus?10. In line 293, "strain has" cannot have extra spaces, please correct it.11.Your manuscript needs to be carefully edited by someone with professional English editing skills, paying special attention to English grammar, spelling, and sentence structure, to ensure that readers have a clear understanding of the research objectives and results
Comments on the Quality of English LanguageSame to suggestions for Authors
Author Response
Response to Reviewer’s Comment: Manuscript ID:viruses-3056036
Original Manuscript Title: Identification and Full-Length Sequence Analysis of a Novel Recombinant Goat Astrovirus Genotype in Guangxi, China
We appreciate your comments on the professionalism of our article, and as you were concerned that there were several issues that needed to be addressed, we have made extensive revisions to the previous manuscript based on your suggestions, with the following corrections:
1.The picture in Figure 2 is not clear, please adjust the picture sharpness.
Response:Sincerely thank you for your valuable suggestions, we have corrected the picture, we hope the revised picture can make you satisfied.
2.Line 86, for "2 × Rapid taq Master Mix", please provide the name of the reagent supplier.
Response:Thank you for your suggestion, we apologize for our carelessness and we have made corrections based on your suggestions, which are noted in red font in the text. See line 92.
3.The paper mentions that different primers were used to amplify the viral genome, but the annealing temperatures used are not seen in Tables 1 and 2.so please add the annealing temperatures of the primers used.
Response:Thank you for your suggestion, which we strongly agree with, and we have corrected Tables 1 and 2 by adding the Tm values of the primers, as suggested by other reviewers Tables 1 and 2 can be found in the Supplementary Section.
4.Line 158, "goat astrovirus GX WZ" should be "goat astrovirus GX WZ 2023", please use the same expression throughout the text.
Response:Thank you very much for your careful reading, and thanks for the reminder that we have made corrections and marked them in red in the text. See line 157.
- Hyperlinks should be added to sequences mentioned in the article to facilitate quick access to sequence information by the reader. For example, line 149, lines 180-185, 236-240.
Response:Thank you for your valuable suggestions, we have made changes by adding hyperlinks to the serial numbers that appear in the article and marking them in bule font in the article.
- The evolutionary tree constructed in the article used fewer reference sequences for ovine, which could be increased.
Response:Thank you very much for your suggestion, we have considered this issue before, but due to the limited reference sequences available for sheep astroviruses based on NCBI at this time, fewer sequences were used in the construction of the evolutionary tree for this study.
7.Lines 43, 158 and 224, "ORF 2" is incorrectly formatted, please check the whole text and correct it uniformly.
Response:We apologize for our carelessness and thank you for the heads up. We have made the correction and marked it in red in the text. See lines 48,157,226.
8.Lines 212 and 216, "s2 m" is incorrectly formatted, please check the whole text and correct it consistently.
Response:Thank you very much for your suggestion, we have made the correction and highlighted it in red in the text. See lines 210, 214.
9.In line 263, the Astv in "AstV was first identified in animals in 1977 in sheep fecal samples" is incorrectly stated, confirming whether it is Astv or sheep astrovirus?
Response:Thank you very much for your suggestion, we apologize for our carelessness and we have made the correction and highlighted it in red in the text, see line 264.
- In line 293, "strain has" must not have a space.
Response:Thank you for your careful scrutiny and we apologize for our carelessness, we have made corrections and highlighted them in red in the text, see line 294.
11.您的稿件需要由具有专业英语编辑能力的人员仔细编辑,特别注意英语语法、拼写和句子结构,以确保读者对研究目的和结果有清晰的认识。
响应: 感谢您的建议,我们已寻求熟悉相关主题英语的人的帮助,以改进文章的语法等。我们希望我们的改进能够让您的期刊满意。

Round 2
Reviewer 1 Report
Comments and Suggestions for Authors
The study of Yulu Zhou and co-workers describes the RT-PCR-based detection and phylogenetic/sequence analyses of two (strain GX WZ 2023 and GX HC 2023) novel astroviruses from goats. The GX HC 2023 strain most likely belongs to a same genotype species (MAstV-2) as its closest relative (caprine AstV GX) while the GX WZ 2023 is most likely a founding member of a novel genotype species due to the low capsid amino acid sequence identity (p-dist >0.338) to any of the known AstVs. The authors also stated that GX WZ 2023 is a recombinant strain.
Since the last review the manuscript was improved in some ways, but still contains multiple flaws and scientific errors (some of them was also part of the original MS), and misleading analyses (see the detailed points below).
Major points
One of the two main problems with the study is (i) that there is still no convincing evidence which support the recombination nature of GX WZ 2023 because the most closely related viral sequences in the ORF2 shows low sequence identity. Therefore, the sections about the recombination analysis should be rephrased more cautiously. I indicated this in my firs review (“GX WZ 2023 could be a recombinant strain, but the sequence similarity between the presumed parental sequences could be the results of adaptive evolutional changes”) but based on the answers of the authors (“ lines: 249 -251: …goat astrovirus GX WZ 2023 may be a recombinant strain, resulting in a change in similarity that may be the result of an adaptive evolutionary change in the sequence of the parental sequence from BAstV 16 2021 CHN.”) I’m afraid they just misunderstood me. I meant that the higher (but still low) sequence identity e.g. in the ORF2 between goat AstV GX WZ 2023 and Bovine AstV 16 2021 CHN could be resulted from the adaptive evolution of goat AstV GX WZ 2023 without any recombination events between sheep/goat or bovine AstVs. The results of an in-silico recombination analysis like RDP4 don’t necessary means that the recombination really had happened and especially not that the recombination was took place between the selected/available AstV sequences. This important uncertainty should be included in all parts of the manuscript (SEE CAPITALS below). E.g.:
In the title “…Novel, POTENTIALLY Recombinant…”
Lines 27-28: “The GX WZ 2023 strain COULD recombined WITH ASTVS RELATED to the Chinese (BAstV 16 2021 CHN) and Japanese bovine strains (BAstV JPN 2015) in the ORF2 region.”
Line 236: “The caprine astrovirus Sichuan strain COULD BE a recombinant with ovine astrovirus S5.1”,
Lines 238-239: “In addition, in another set of recombination events, we found BAstV 16 2021 CHN strain COULD be recombinant…”
Lines 242-243: “It was found that all of the above recombination events COLULD occurred within the ORF2.”
Line 255: “…astroviruses that COULD have recombined with bovine astrovirus….”
Lines 308 - 310: “In this study, we unexpectedly found that goat astrovirus GX WZ 2023 COULD recombined with ovine astrovirus S5.1 and caprine astrovirus Sichuan strains in the ORF2 region.
Lines 324-325: “This study identified for the first time TRACES OF POTENTIAL recombination events of astroviruses within a goat population as well as between goats and cattle in Guangxi…”
(ii) The second main problem is that the authors state that “The full-length sequences of GX HC 2023 and GX WZ 2023 were 6213 nt and 6284 nt, respectively” (in lines 125-126 of the original MS), but according to the GenBank flatfile of GX HC 2023 (PP571882), although the sequence is indeed 6213-nt-long, but far from complete, it’s 5’ end is still missing, including the 5’ UTR and the start of ORF1a. I don’t know where the authors got the information about the organization of the 5’ end of the genome of GX HC 2023 in figure 1? Furthermore, the MW784098 is an ovine AstV not caprine.
The GenBank flatfile of GX HC 2023 (PP571882) is still indicates that this sequence is “partial”, please update it. The Figure 1 is still the same as in the original MS (based on the GenBank MW784098 is an ovine/sheep AstV NOT a goat). Most of the numbers of “Caprine astrovirus GX” and “Goat Astrovirus GX” in figure 1 are the same, why, is it correct? Methods and primers of the 5’ end PCR should be included to the MS/supplementary.
Details of cloning and sequencing of the PCR products as well as the methods used for the determination of the 5’ and 3’ ends are still completely missing from the MS.
Minor points
Lines 24, 26, 151, 153, 292: The authors are incorrectly using the term “homologies” throughout the entire manuscript. I think the term “identities” is more appropriate in case of sequence comparisons.
Line 234: What does it mean “chromosome” ?
Line 241: What does it mean “goat aspergillus virus GX”?
Lines 262-263: The us of the term “two new astrovirus species” here is incorrect!
Line 299: Please change “genotypic species” to “genotype species”
Figure 4: The resolution of the image is STILL too low (the results should be imported to a Vector Graphics Software like Corell Draw).
Author Response
Response to Reviewer’s Comment 1: Manuscript ID:viruses-3056036
Original Manuscript Title: Identification and Full-Length Sequence
Analysis of a Novel Recombinant Goat Astrovirus Genotype in Guangxi, China
We faithfully thank the editors and all reviewers for their valuable suggestions! As you are concerned, there are several issues that need to be addressed. Based on your suggestions, we have revised and corrected our previous manuscript with the following corrections:
Specific Comments
- there is still no convincing evidence which support the recombination nature of GX WZ 2023 because the most closely related viral sequences in the ORF2 shows low sequence identity. Therefore, the sections about the recombination analysis should be rephrased more cautiously. I indicated this in my firs review (“GX WZ 2023 could be a recombinant strain, but the sequence similarity between the presumed parental sequences could be the results of adaptive evolutional changes”) but based on the answers of the authors (“ lines: 249 -251: …goat astrovirus GX WZ 2023 may be a recombinant strain, resulting in a change in similarity that may be the result of an adaptive evolutionary change in the sequence of the parental sequence from BAstV 16 2021 CHN.”) I’m afraid they just misunderstood me. I meant that the higher (but still low) sequence identity e.g. in the ORF2 between goat AstV GX WZ 2023 and Bovine AstV 16 2021 CHN could be resulted from the adaptive evolution of goat AstV GX WZ 2023 without any recombination events between sheep/goat or bovine AstVs. The results of an in-silico recombination analysis like RDP4 don’t necessary means that the recombination really had happened and especially not that the recombination was took place between the selected/available AstV sequences. This important uncertainty should be included in all parts of the manuscript (SEE CAPITALS below).
Response: Thank you very much for your careful reading, and we have given careful consideration to the fact that you have made a suggestion, and I did misunderstand what you meant about my first response. But we still think that the reorganization of GX WZ 2023.
First of all: probably because the WZ strain is a new genotype in goat astrovirus, so the most closely related viral sequences in ORF2 in GX WZ 2023 show low sequence identity.
Secondly, for the recombination analysis in the article, we have also referred to several papers, such as the article “Identification of a novel astrovirus in goats in China”, DOI: 10.1016/j.meegid.2021.105105 . the analysis results of this article, SWUN/ F4/2019 strain is also a new genotype, SWUN/ F4/2019 and CapAstV-G5.1 nucleotide sequence identity is the highest (77.0%), the closest relatives, but also using the RDP 4.0 software to recombination analysis of the sequence, and found that SWUN/ F4/2019 recombined with CapAstV-G5.1;
Third, the article “Diversity of Astrovirus in Goats in Southwest China and Identification of Two Novel Caprine Astroviruses” DOI: 10.1128/spectrum.01218-22. In this article, RDP 4.0 software was also used to analyze the recombination of sequences, and it was found that recombination occurred between goats and goats, and these recombinations also occurred between selected astrovirus sequences.
Finally, in lines 281-284 of this article, recombination has also been reported to occur between astroviruses of different species. In summary, so we think that the recombination events predicted in this article using the RDP 4.0 software are valid. I have also made a new correction to the analysis of the recombination part, see lines 229-235. and I thank you again for your suggestion, and I hope that my correction this time will satisfy you.
2.The second main problem is that the authors state that “The full-length sequences of GX HC 2023 and GX WZ 2023 were 6213 nt and 6284 nt, respectively” (in lines 125-126 of the original MS), but according to the GenBank flatfile of GX HC 2023 (PP571882), although the sequence is indeed 6213-nt-long, but far from complete, it’s 5’ end is still missing, including the 5’ UTR and the start of ORF1a. I don’t know where the authors got the information about the organization of the 5’ end of the genome of GX HC 2023 in figure 1? Furthermore, the MW784098 is an ovine AstV not caprine.
The GenBank flatfile of GX HC 2023 (PP571882) is still indicates that this sequence is “partial”, please update it. The Figure 1 is still the same as in the original MS (based on the GenBank MW784098 is an ovine/sheep AstV NOT a goat). Most of the numbers of “Caprine astrovirus GX” and “Goat Astrovirus GX” in figure 1 are the same, why, is it correct? Methods and primers of the 5’ end PCR should be included to the MS/supplementary.
Details of cloning and sequencing of the PCR products as well as the methods used for the determination of the 5’ and 3’ ends are still completely missing from the MS.
Response: Thank you very much for your careful reading and I apologize for my mistake. I double and triple checked the GenBank planar file for GX HC 2023 and it was indeed my mistake, it did not show the 5` UTR on it. it was my misclassification of the start codon region of ORF1a that led me to think that my sequence contained the 5` UTR region, whereas GX HC 2023 is indeed part of the genome, it is still missing the 5' end, both the 5' UTR and the the start of ORF1a. I really apologize for this for my mistake. I have made a few corrections:
First, I corrected the genome description of GX HC 2023 in the article, which also includes the reconstruction of the ORF1a amino acid evolutionary tree, and the recalculation of the 0RF1a amino acid identity in Tables 3 and 4.
In addition, by searching the GenBank file and the original article of MW784098, and according to your suggestion, I have also corrected the nomenclature of MW784098, and changed caprine astrovirus GX to ovine astrovirus GX, which includes all the pictures and the article about caprine astrovirus GX (MW784098). and once again, I apologize for my carelessness.
Third, The reason why most of the numbers of “caprine astrovirus GX” and “goat astrovirus GX HC” are the same in Figure 1 is that “caprine astrovirus GX” has the highest nucleotide identity with “goat astrovirus GX HC”. I used “caprine astrovirus GX” as the reference sequence to amplify “goat astrovirus GX HC”, so most of the numbers are the same. However, because I had misclassified the start codon region of ORF1a, I found that both “goat astrovirus GX HC” and “caprine astrovirus GX” do not contain the 5 `UTR region, so I corrected Figure 1 as well.
Finally, the details of the cloning and sequencing of the PCR products are in lines 97-99 and 92-93 of the article, respectively; and regarding the assays, they were amplified with primers designed according to the 5' and 3' ends of the sequences of the reference strains on NCBI. For example, the GX WZ strain was amplified by designing primers based on the 5' and 3' ends of Ovine astrovirus S5.1 and Ovine astrovirus S6.1, so the primers for amplification of the 5' and 3' ends have been included in the primers in Table 1 and Table 2. Corrections to the above issues are highlighted in red in the article. I sincerely apologize for my mistake and hope my improvement will satisfy you.
3.Lines 24, 26, 151, 153, 292: The authors are incorrectly using the term “homologies” throughout the entire manuscript. I think the term “identities” is more appropriate in case of sequence comparisons.
Response: Thank you very much for your careful reading, and I apologize for our carelessness. The word “homologies” has been replaced with “identities” and is highlighted in red throughout the article.
4.Line 234: What does it mean “chromosome” ?
Response: Thank you very much for your careful reading. The word “chromosome” was a misrepresentation on my part, I meant to express the abscissa axis, which I have corrected in the text and highlighted in red. See line 219.
5.Line 241: What does it mean “goat aspergillus virus GX”?
Response: Thank you very much for your careful reading, and I apologize for our carelessness. The term “goat aspergillus virus GX” has been changed to “goat astrovirus GX” and is marked in red in the article.
6.Lines 262-263: The us of the term “two new astrovirus species” here is incorrect!
Response:Thank you very much for your suggestion, I have made the correct corrections in the article and highlighted them in red in the text, see line 245. I hope we can make the changes to your satisfaction!
7.Line 299: Please change “genotypic species” to “genotype species”
Response:Thank you very much for your careful reading, I have made the correct corrections and highlighted them in red in the article, see line 278.
8.Figure 4: The resolution of the image is STILL too low (the results should be imported to a Vector Graphics Software like Corell Draw).
Response:Thank you very much for your suggestion, I have re-adjusted the resolution of the images for a better feel for the readers, I hope this correction will satisfy you.
